# Exogenous Contrast Agents in Photoacoustic Imaging: An In Vivo Review for Tumor Imaging

**DOI:** 10.3390/nano12030393

**Published:** 2022-01-25

**Authors:** Afifa Farooq, Shafiya Sabah, Salam Dhou, Nour Alsawaftah, Ghaleb Husseini

**Affiliations:** 1Biomedical Engineering Graduate Program, American University of Sharjah, Sharjah 26666, United Arab Emirates; g00051241@alumni.aus.edu (A.F.); g00079837@alumni.aus.edu (S.S.); nalsawaftah@aus.edu (N.A.); 2Department of Computer Science and Engineering, American University of Sharjah, Sharjah 26666, United Arab Emirates; 3Department of Chemical Engineering, American University of Sharjah, Sharjah 26666, United Arab Emirates

**Keywords:** photoacoustic imaging, exogenous contrast agents, tumor imaging

## Abstract

The field of cancer theranostics has grown rapidly in the past decade and innovative ‘biosmart’ theranostic materials are being synthesized and studied to combat the fast growth of cancer metastases. While current state-of-the-art oncology imaging techniques have decreased mortality rates, patients still face a diminished quality of life due to treatment. Therefore, improved diagnostics are needed to define in vivo tumor growths on a molecular level to achieve image-guided therapies and tailored dosage needs. This review summarizes in vivo studies that utilize contrast agents within the field of photoacoustic imaging—a relatively new imaging modality—for tumor detection, with a special focus on imaging and transducer parameters. This paper also details the different types of contrast agents used in this novel diagnostic field, i.e., organic-based, metal/inorganic-based, and dye-based contrast agents. We conclude this review by discussing the challenges and future direction of photoacoustic imaging.

## 1. Introduction

Photoacoustic imaging (PAI) [1] is a breakthrough invention in the field of bioengineering [2]. It involves the transmission of a short pulse of laser non-invasively onto the tissues. The absorbed light changes into heat, causing the thermal excitation and expansion of the tissues. This results in generating acoustic waves that can be detected via a transducer [3]. PAI offers several advantages, including the utilization of non-ionizing radiation, the ability to reach greater depths, as well as increased resolution and optical contrast [4]. The discovery of the photoacoustic (PA) effect dates back to 1880, when the phenomenon was first observed by Alexander Graham Bell [5]. Bell reported that sound is produced upon the illumination of solid materials with a light beam. The light beam must be of a vibratory nature for the PA effect to be observed. As laser technology became available in the 1960s, the PA effect began to be considered for practical applications. By the 1990s, researchers working in the field recognized the potential use of the PA effect for medical imaging [6,7,8,9]. R. Esenaliev et al. [6] studied the feasibility of using optoacoustic laser imaging for the early detection of breast cancer. In the phantom study performed in [6], it was observed that in comparison to ultrasound (US) and mammography, the new laser optoacoustic imaging technique offers images with improved contrast for phantoms that were dense and uniform acoustically. Over time, with improvements in hybrid imaging technologies and advancements in image reconstruction algorithms, it became possible to use the PA effect for in vivo studies [10,11,12]. In the past few years, a significant evolution in PAI is evident, which will be further discussed in this paper with a focus on contrast agents used in in vivo PAI studies (refer to Figure 1).

Materials that absorb the transmitted optical signal can be endogenous, i.e., present naturally in the body, or exogenous, i.e., synthesized/developed outside the body. In vivo studies have been performed to detect tumors using probes that can be activated by enzymes related to cancer [13], probes based on peptides [14], gold nanoparticles [15,16], dyes [17,18,19], melanin, oxy- and deoxy-hemoglobin [20,21]. Previous reviews have summarized methods that utilize different contrast agents. However, these publications fall short of presenting the different agents based on image quality, blood circulation time, cytotoxicity levels, and laser pulse parameters. Table 1 summarizes contrast agents used for various imaging modalities.

As PAI is a low-cost, non-invasive imaging modality that uses non-ionizing laser power for the excitation of target tissues; it can provide real-time physiological information such as blood oxygenation and other structural information of the site being studied [22]. The penetration depth offered by PAI is higher than that conventionally offered by optical imaging. Apart from providing great benefits, PAI has certain limitations. Optical fluence, i.e., the deposited optical energy on the target region through laser, is depth-dependent and can lead to lower amplitudes of the PA signal arising from deep tissues and vessels [23,24]. Therefore, achieving a clinically desired penetration depth with PAI is challenging, thus limiting its use in widespread medical applications. Another challenge with PAI occurs while imaging regions with overlapping tissue types as the acoustic properties between different tissue types are not consistent [25]. In addition, the safety and compatibility of exogenous contrast agents used in PAI with the biological system need to be ensured.

Due to the rising trend of imaging tumor tissues using PAI [26,27,28,29,30,31,32,33,34] (see Figure 2), this review summarizes the recent major developments in PAI. It summarizes the in vivo studies that utilize different contrast agents to produce an enhanced image. Furthermore, this review aims to classify these contrast agents into organic-based, metal-based, and dye-based agents with a detailed commentary on significant points of interest. The rest of the paper is organized as follows. Section 2 discussed the contrast agents used for in vivo studies. Section 3 concludes the paper and presents the challenges facing PAI along with future research directions.

**Table 1 nanomaterials-12-00393-t001:** Contrast agents used for different imaging modalities.

Modality	Commonly Used Contrast Agents	References
**MRI**	Gadolinium, Super paramagnetic iron oxide nanoparticles (SPIONs), Carbon-13, Nanodiamonds, Carbon nanotubes, Graphene, Manganese, Silicon, Peptides	[35]
**CT**	Gold nanoparticles, Iodine (131I), Bismuth, Lathanide-based (gadolinium, dysprosium, ytterbium)	[35,36]
**Ultrasound**	Nanobubbles, microbubbles (with modifications)	[35,37]
**PET**	Gold nanoparticles, Copper (64Cu), Iodine (124I), Fluorine (18F)	[35,38]
**SPECT**	Gold nanoparticles, Technetium (99mTc)	[35]
**Optical Imaging**	Fluorescence, Quantum dots, Gold nanoparticles, Persistent luminescence nanoparticles	[35]
Combinations of these contrast agents can be used to create hybrid contrast agents and optimize imaging		[38,39]

## 2. Contrast Agents for In Vivo Testing

### 2.1. Basis of PAI and Design Considerations for Contrast Agents

PAI is based on PA tomography (PAT) that incorporates optical imaging and US. A short pulse of laser is transmitted onto/into the tissues non-invasively. The absorbed light changes into heat, causing the thermal excitation and expansion of tissues [40]. This results in generating acoustic waves that can be detected via a transducer. One of the principal subdivisions in PAT is photoacoustic microscopy (PAM). In PAM, every excitation by a laser pulse creates an image in one dimension of a point in the object. Numerous one-dimension images can be combined to form a three-dimension image without using reconstruction algorithms [3].

Endogenous and exogenous contrast agents are used in PAI. Endogenous molecules found in the body, such as melanin and hemoglobin, have broad optical absorption in the visible light and near-infrared (NIR) ranges compared to adjacent tissues, leading to strong PA contrast signals. However, information obtained by imaging a tumor in its initial phase using endogenous contrast agents is not adequate; hence, biocompatible exogenous contrast agents are used. The latter provides an enhanced signal contrast and better image quality compared to endogenous contrast agents [41].

Contrast agents in PAI are usually chosen to optimize absorption and depth in vivo. For this reason, contrast agent wavelengths should be in the NIR range. There is minimal light absorption in this region by hemoglobin above 650 nm, and by water below 900 nm [42]. More recently, the NIR-II window, in the 1000–1700 nm range, is a focus of contrast agent development as it has a greater tissue penetration depth due to lower light-tissue interaction and safe use of higher power density for light irradiation [43]. Longer penetration depth and higher contrast provided by NIR-II window compared to NIR-I window are because of reduced tissue scattering and minimal tissue absorption [44]. In short, minimal absorption means more significant penetration depths. Some of the major factors that must be taken into consideration to optimize imaging in vivo are: (1) Having long blood circulation times, i.e., these agents or nanoparticles should be able to evade the immune response by the reticuloendothelial system (RES) cells, travel across membrane barriers, and avoid elimination by renal and splenic filtration and reach the imaged tissue, (2) Being biocompatible, i.e., not cytotoxic within the concentration dosage to be administered for imaging, (3) Exhibiting a strong imaging signal to improve poor contrast, (4) Possessing biological specificity, i.e., being designed such that it can reach the imaged area in a reasonable time frame [42].

Since this review will mainly focus on imaging cancer malignancies, it is necessary to mention the three methods of tumor targeting in oncology: passive, active, and triggered targeting. Passive targeting involves the utilization of the enhanced permeability and retention (EPR) phenomenon, which increases the accumulation of nanoparticles (in the size range of 15–150 nm) at the tumor site [45]. This accumulation occurs due to the presence of underdeveloped blood vessels at tumor sites, with fenestrations and haphazard blood flow. Furthermore, due to the fast-growing nature of cancerous tissue, there are few to no lymphatic vessels, and thus, low drainage allows a 10–50 times higher accumulation than healthy tissues [46]. As mentioned earlier, several factors should be taken into consideration when designing these nanocarriers. The optimal size, 10–150 nm, is an essential factor to avoid filtration by the kidneys, and/or uptake by the liver- unless the region of interest includes these organs [45]. Active- or ligand- targeting uses receptor-mediated endocytosis to internalize nanocarriers. The surface of these drug encapsulating vehicles is decorated with targeting moieties (e.g., folic acid, estrone, Herceptin, etc.) that direct the probe to receptors overexpressed on the surface of the cancer cells. Triggered targeting can be achieved using internal and external means. Internal triggers include temperature, enzyme concentration, and pH levels, while external stimuli include US, NIR, and electromagnetic waves. As the performance of PAI is dependent on optically absorbing components within synthesized nanostructures, a combination of these components generally leads to better results. Therefore, this paper aims to broadly classify these multimodal/hybrid nanoagents based on component materials with PA functionality (while other components in the nanoagents perform either therapeutic, other imaging types or biocompatibility functions).

### 2.2. Organic Contrast Agents

Since each class of materials has specific advantages for different applications, the key advantage of organic contrast agents is that they are non-toxic and can degrade in living bodies. Nanosystems are the focus of current diagnostic, theranostic and drug delivery research due to the versatility of these materials. They are increasingly combined with organics nanoparticles and biodegradable polymers to build a new generation of contrast agents that enhance specific parameters. In addition, their size, shape, mechanical flexibility and surface chemistry can be modified to optimize their efficacy in clinical applications [47,48,49]. Due to their in vivo high biocompatibility, fluorescent nature, and modification flexibility, Xiao et al. [50] synthesized and characterized melanin carbonaceous dots (MCDs) for dual PAI and fluorescence imaging of breast cancer tumors in mice. With MCDs having an absorbance peak at 633 nm, near the infrared lower wavelength limit of 700 nm, higher spatial resolution imaging was produced for higher penetration depths. MCDs, for PAI, were combined with fluorescent imaging to improve the low spatial resolution of the latter. The major advantages of MCDs are their (1) prolonged blood circulation time, and (2) in vivo accumulation in triple-negative breast cancer (4T1) xenografts. However, more work is needed before these probes proceed to clinical trials, as their overall toxicity was high. In addition to tumors, they also accumulated heavily in the liver and kidneys and can therefore, be used to image both organs.

However, due to the tedious, costly, and complex synthesis of carbon dots (CDs), research groups have been extracting/deriving these CDs from natural resources such as *Hypocrella bambusae* (a parasitic fungus) [51], ethylene diamine, phosphoric acid and citric acid [52], and polythiophene phenylpropionic acid (refer to Figure 3) [53]. The enzyme horseradish peroxidase, derived from horseradish roots, was also used in conjunction with 2,2′-azino-bis(3-ethylbenzothiazoline-6-sulfonic acid) (ABTS) and encapsulated in liposomes to act as an H_2_O_2_-responsive in vivo nanoprobe [54]. The idea behind these nanoprobes is that they will react directly with the targeted molecule and thus, exhibit high specificity to the desired molecule.

The nanoprobes mentioned above were used in cancer theranostics to detect H_2_O_2_ levels via the following method: synthesized glucose oxidase PEGylated (polyethylene glycol, PEG) liposomes, for starvation therapy in a 4T1-tumor microenvironment, were metabolized by the cancer cells to form gluconic acid and H_2_O_2_. The synthesized nanoprobe then detected the H_2_O_2_, thus, implementing an innovative theranostic strategy [55].

Since a major branch of theranostics in cancer nanomedicine deals with lipids for drug encapsulation, contrast agent encapsulation, ligand targeting, and biocompatibility coating, it is necessary to discuss polymeric nanoparticles and their role as contrast agents in PAI. Here it becomes essential to remind the reader that lipids exhibit optical absorbance but to a lesser extent compared to other endogenous contrast agents. Consequently, Liu et al. [56] synthesized two novel triblock copolymer nanosystems, poly(2-methyl-2-oxazoline)-block-poly (dimethylsiloxane)-block-poly(2-methyl-2-oxazoline) (PMOXA-b-PDMS-b-PMOXA); one with positively-charged terminal amino acid groups (P-NPs) and the other with negatively-charged terminal carboxylic groups (N-NPs). These micelles, spherical self-assembling single-layered nanocarriers with a hydrophobic core, were used to encapsulate a hydrophobic NIR photonic agent called hydrophobized phthalocyanine Zinc complex (H-PcZn) and the absorbance peak of both nanosystems was measured to be 680 nm—note that this is within the NIR window. A linear increase in the intensity of the PA signal was also observed with a logarithmic increase in the concentration of both P-NPs and N-NPs. The PA signal plateaued within 10 min of in vivo administration in mice, with an increased accumulation observed in the spleen compared to the liver and kidneys for N-NPs, whereas P-NPs exhibited increased accumulation in both the liver and spleen as compared to the kidneys and a comparatively stronger signal (by a factor of 1.5 in the spleen) in other tissues. Therefore, this signifies that organ-specific applications for each nanosystem can be developed. Still, longer incubation times are required to measure blood circulation times and the use of targeting moieties for specific tissues and tumors to enhance tumor targeting abilities [56,57] (see Table 2). In general, the advantage of using conjugated polymers, aside from their NIR absorbance/emission for PAI and photothermal therapy, is the ability to modify side chains to achieve desired characteristics such as self-assembly, increased protein corona formation for better nanoparticle biodistribution and targeted applications with the band gap in their π-π* electronic transition influencing emitted spectra and leading to tunable optical properties [58,59].

A problem associated with using US transducers is signal intensity loss due to US transducers having limited frequency range detection; therefore, a study by Kim et al. [60] used porphyrin phospholipid microbubble to produce resonance-based frequency-selective PA signals. Porhyrin, and other small-molecule organic dyes such as cyanine and squaraine, are being used increasingly due to their flexibility for structural modification and faster clearance with porphyrin being used as contrast and phase-change from nanodroplets to microbubbles for both ultrasound and PAI [61,62]. Theranostic molecular engineered dyes such as heptacyclic B, O-chelated BODIPY organic molecules have also demonstrated 14 times enhancement in PAI contrast post 10 h injection along with 58.7% photothermal conversion for hyperthermic tumor ablation [63]. Development of self-assembling pH-sensitive charge-transfer nanocomplexes by 3,3′,5,5′-tetramethylbenzidine within the NIR-II window has shown good contrast up to 5 cm and can be further investigated for NIR-II PA contrast agents [64]. Similarly, Pu et al. [65] used the electronic and optical properties of semiconducting oligomers and BODIPY dyes. The former acted as a PA matrix and the latter acted as PA signal enhancer and pH indicator for improved ratiometric response and increased pH sensitivity. Another study used oligomerization to its advantage for PA signal modulation based on intramolecular interactions [66]. Recent reviews summarizing studies on organic contrast agents and their future trends demonstrate a greater shift towards contrast agents in the NIR-II window [33,67]. Further studies were detailed in Table 3 with additional studies found in [68,69,70,71,72,73,74,75,76,77,78,79,80,81,82,83,84,85,86,87,88,89,90,91].

**Table 3 nanomaterials-12-00393-t003:** Classification and Summary of Tumor-Targeting In Vivo Studies for PAI Organic Nanoparticles/ Nanosystems Contrast Agents.

Classification	Material Used	Imaging Modalities	Application	Studies Conducted	Relevant Measured Parameters	Transducer Used	Computational Techniques	Publication Year/Reference
Semiconducting Polymer	poly{3-(5-(9-hexyl9-octyl-9H-fluoren-2-yl)thiophen-2-yl)-2,5-bis(2-hexyldecyl)-6- (thiophen-2-yl)pyrrolo [3,4-c]pyrrole-1,4(2H,5H)-dione} (PDPPF, SP0) with SP5 and SP10 (self-quenching SPs)	PAI	Imaging of breast cancer and cervical cancer tumors	HeLa cervical adenocarcinoma epithelial cells for In vitro; In vivo and ex vivo on 4T1 breast cancer tumor in mice/mice organs	Maximum PA signal of SP10 at 4h for both SP10-RGD and SP10 with slower clearance rate for SP10-RGD and 1.78 fold higher PA intensity for SP10-RGD as well	LAZR instrument (Visualsonics, 2100 High-Resolution Imaging System)	-	2017 [92]
Derived from natural resources	DPAHB nanovesicles (hypocrellin B (HB) modified with 1,2-diamino-2-methyl propane encapsulated by PLGA-PEG)	PAI, fluorescence, photodynamic and photothermal therapy	Imaging of 4T1 breast cancer tumors	In vitro and in vivo PAI.	High-intensity signals and enhanced spatial resolution was achieved using DPAHB nanovesicles. PA signal intensity attained maximum peak at 12 h after injection of nanovesicles.	MSOT inVision 128 PAT system		2018 [93]
	Other							2018 [51] 2017 [52,54] 2015 [53]
Carbon nanodots	Nitrogen-Doped Carbon Nanodots	PAI	Imaging of sentinel lymph node to detect metastatic cancer	In vivo and ex vivo mapping of sentinel lymph node, in vivo PAI of the bladder.	Post injecting N-CNDs PA signal reached a peak at 30 min, and the signal kept decreasing until 180 min. Results show that the contrast agent was circulating in the lymphatic system before being degraded.	Ultrasound transducer with spherical focusing and having a 5-MHz central frequency, Acoustic-resolution reflection-mode PA imaging system	Raster scanning to acquire PA images	2016 [94]
Organic small molecule	Diradicaloid molecular (DRM) structure	PAI and PTT	Imaging of A549 lung cancer	PAI-guided PTT in vitro and in vivo	The average PA signal of tumors excised from the mice injected with DRM NPs is over 4 times higher than that from the control group	Vevo LAZR-X imaging equipment	DFT calculations of optimized geometries of the DRM in the ground and excited states	2021 [95]
Mitochondria-targeted BODIPY NPs	BODIPY NPs with a cationic triphenylphosphine (TPP) group (Mito-BDP1–5 NPs) bearing different lengths of ethylene glycol (0–4 units), along with HO-BDP5 without a cationic TPP group	PAI and PTI	Imaging of mitochondria in HeLa cells	In vitro mitochondrial imaging, and in vivo PTI and PAI	Mito-BDP5 possessed high photothermal conversion efficiency (η) of 76.6%, and was able to accumulate in the tumor sites through the EPR effect, subsequently strong PT and PA signals can be observed in tumor sites.	PAI was conducted on a PA computed-tomography system equipped with a 10 MHz, 10 mJ cm^−2^, 384-element ring ultrasound array, and a tunable pulsed laser	-	2021 [96]
Carbon nanohorns	carbon nanohorn-polyglycerol-gold (CNH-PG-Au) NPs	PAI and x-ray	Imaging of 4T1 mouse breast cancer cells	In vivo PAI of tumor treatment using DOX@CNH-PG-Au	The photoacoustic intensity of the tumor site increased gradually and reached a maximum 48 h post-injection (735 ± 47), indicating that DOX@CNH-PG-Au NPs steadily accumulated in the tumor during this period	MSOT inVision 256 PAI systems	-	2021 [97]
Laponite (LAP) nanoplatforms	polydopamine (PDA) coated LAP nanoplatforms modified with polyethylene glycol-arginine-glycine-aspartic acid (PEG-RGD)	PAI	Imaging of 4T1 mouse breast cancer cells	In vitro and in vivo PAI-guided chemo-phototherapy of cancer cells	NPs showed an increased PA signal at tumor sites after injection, and the PA signal peaked at 2 h post-injection.	Vevo LAZR PAI system equipped with an 875 nm laser	-	2021 [98]

### 2.3. Metal/Inorganic Contrast Agents

Metals such as gold and silver significantly enhance contrast in PAI as their optical absorbance is based on surface plasmon resonance (SPR). Therefore, these structures exhibit higher absorbance than other optical agents such as dyes [26]. Metallic agents can also serve as contrast agents in multiple imaging modalities, and so can inorganic nanoparticles. There is increasing interest in using inorganic contrast agents as desirable properties such as tunable peaks in the NIR regions, better brightness, superior photostability, and magnetic and optical scattering and absorption, along with luminescence, renders them useful for multimodal imaging purposes [99]. Ma et al. [14] also synthesized a biocompatible nanosystem that conjugated PEGylated gold nanorods to oligopeptides PT6 and PT7, termed PGNR-PT6 and PGNR-PT7, respectively. These actively targeted nanosystems exhibited an absorbance peak at 810 nm and a high specificity towards osteosarcoma in UMR-106 tumor-bearing mice. Furthermore, a remarkable 2.6- and 3.4-fold contrast from PGNR-PT6 and PGNR-PT7, respectively, was found compared to a PBS-administered control group. Due to the reasons mentioned above and their low cytotoxicity, these nanorods have a strong potential for osteosarcoma-PAI applications. Another study conducted by Li et al. [15] incorporated a liquid perfluorocarbon (perfluorinated hexane/PFH), used as a contrast agent in cancer biomedicine, and gold nanorods, used in PAI due to its strong light absorption in the NIR window, with the nanoemulsion encapsulated in a poly(lactide-co-glycolide) (PLGA) shell. This highly biocompatible shell was then conjugated to a monoclonal MAGE-1 antibody to target melanoma cancer therapy specifically by binding to melanoma antigens. This nanosystem (MAGE-Au-PFH-NP) can be used in dual-modality contrast agents for PA and US imaging. With an encapsulation efficiency of 70.61%, the MAGE-Au-PFH-NPs showed an enhanced PA signal and a linear increase in absorption intensity and concentration. In vivo experiments using melanoma-bearing mice demonstrated peak PA signals after 2 h of injection and a liquid-gas phase transformation after laser irradiation to form microbubbles, functioning as efficient US contrast agents, with a final temperature increase to 70 °C (of the gold nanorods), leading to a PFH phase change. The US signals reached peak values 10 min after laser irradiation and therefore, this nanosystem is a suitable contrast and treatment agent for melanomas. While gold-based contrast agents are a popular choice in PAs, it has several drawbacks, including photoinstability and difficulty in synthesis at the nanoscale [100].

Due to the high optical absorption coefficient of blood in the NIR window, researchers explored the use of magnetic nanoparticles (MNPs) as PA contrast agents, where the magnets’ motion directly influences signal intensity, leading to high contrast images with suppressed images background signals [100,101]. Chau et al. [102] reviewed magnetic molecular imaging and targeting for carbon nanomaterials. They reported that prior to 2015, researchers were functionalizing carbon nanotubes, plated with gold, with proteins for targeting and iron oxide cores for magnetic manipulation. Yang et al. [103] employed the magnetic properties of metals for the trimodal imaging (PET/MRI/PAI) of tumor tissue. Apoferritin, an unloaded natural iron protein with a cage-like structure and transferrin receptor 1 (TfR1) targeting ability, was loaded with melanin nanoparticles (AMF), 64Cu2+ and Fe3+ to image HT29, high TfR1 expression, and HepG2, low TfR1 expression, in tumor-bearing mice. Not only were AMF NPs found to be well suited to the PET and MRI imaging modalities, but the PA signal was twice that of the controls used in the HT29 mice model. Due to the low TfR1 expression in the HEPG2 mice models, there was no obvious contrast observed. To enable the clinical translation of MNPs to the field of PAI Li et al. [100] not only synthesized a MNP with a functionalized folic acid PEGylated polypyrrole (PPy) shell by polymerizing PPy on the MNP surface to target cervical tumors in a mouse model, but also developed a second-generation magento-motive PAI (mmPAI) system to reduce background noise produced from static and moving tissue incorporating cyclic magnetic motion and US speckle tracking (see Figure 4). A major point of interest in this study was that since conventional PAI cannot easily distinguish a tumor unless there is already prior knowledge of the tumor location, mmPAI was employed to improve the contrast by two orders of magnitude compared to conventional PA images. This imaging technique can detect 100 to 100,000 tumor cells, whereas current technology has the ability to detect tumor sizes in the mm^3^ to cm^3^ range [100]. Other groups such as Wu et al. [104] and Chang et al. [105] have also utilized the magnetic properties of metals, gadolinium/bismuth, and manganese tungsten oxide, respectively, to provide simultaneous multimodal imaging and therapy for oncological purposes (see Table 2). Additionally, studies using low-frequency range PAI to track metallic nanoparticles and their aggregates, in this case, silver nanoparticles and zinc oxide, are available and can be further used to monitor individual scattering and absorption, specifically in the presence of multiple contrast agents [106,107].

The advent of theranostics, the combination of diagnostics and therapeutics, led to the development of several nanoparticles that functioned both as multimodal imaging agents and had a therapeutic function. Hence, novel nanosystems, based on previously well-researched agents, were resynthesized, incorporating desirable characteristics to produce optimal nanosystems suitable for theranostics. Challenges such as complex synthesis routes, decreased biocompatibility, and high instability became an issue [108]. Gold nanoparticles, among the most widely researched and promising exogenous contrast agents [109], were synthesized as gold nanoclusters with strands of polyallylamine and their surface-functionalized with bovine serum albumin (BSA) to serve as diagnostic in vivo PA contrast agents as well as hyperthermic-anticancer agents—for 4T1 tumor-bearing mice [110]. A study using ultra-small Cu_2_ZnSnS_4_ (CZTS) nanocrystals functionalized with BSA combined the high NIR photothermal ability of the crystals to simultaneous provide photothermal therapy (PTT), along with dual PA/MRI for in vivo H22 liver tumor-bearing mice where the accumulation of the nanosystem and therefore, the signal intensity was found adequate for image-guided tumor therapy [108]. Antimony (Sb) was mentioned by Li et al. [111] as a potential contrast agent in the field of PAI, and it was further researched by Hou et al. [112] by preparing an oleylamine coated Copper-Antimony- Sulfur (Cu-Sb-S) nanoparticle functionalized with poly(vinylpyrrolidone) (PVP) to act as a PA contrast agent and for use in PTT/photodynamic therapy (PDT). Hong et al. [113] demonstrated that PEGylated melanin dots loaded with gadolinium could serve as effective tumor contrast agents in MRI, and later the same group investigated this agent in vivo [114]. Additional studies were conducted in [75,76,77,79,110,115,116,117,118,119,120,121,122] (2018), [114,123,124,125,126] (2017), [127,128,129,130,131,132,133] (2016), [111,134,135,136] (2015) with further detailed studies summarized and classified in Table 4. In addition, Figure 5 depicts an example of the role contrast agents play in indicating tumor sites.

### 2.4. Dye-Based Contrast Agents

Studies have shown the ability of some exogenous dyes to absorb NIR light and produce high-intensity PA signals leading to enhanced image contrast [74,115,137,138] (2018); [85,139,140,141] (2017); [142,143,144,145,146,147] (2016); [80,88,148,149,150,151,152,153] (2015). One of the most commonly used dyes is indocyanine green (ICG) that is FDA approved. ICG is typically used in combination with other materials to attain a maximum PA signal. G. Wang et al. [142] used a nanocomplex consisting of hyaluronic acid and ICG enclosed within a nanotube of carbon to detect squamous cell carcinoma in mice. J. Chen et al. [143] detected Hela xenografts in mice using contrast agents made of ICG, polyethylene and nano-graphene oxide. The results showed that the composite could target the tumor passively and had a prolonged circulation time. Gao et al. [144] combined ICG with superparamagetic iron oxide for the in vivo detection of tumor imaging brain vasculature. In addition to ICG, other dyes such as NIR dyes (IR-825) [85,148], the black hole quencher (BHQ) dye [139], and DiR fluorescent dyes [145] have been used extensively for PAI. A detailed description of dye-based contrast agents and their application are listed in Table 5.

**Table 4 nanomaterials-12-00393-t004:** Classification and Summary of Tumor-Targeting In Vivo Studies for PAI Metal-based/ Inorganic Contrast Agents.

Classification	Material Used	Imaging Modalities	Application	Studies Conducted	Relevant Measured Parameters	Transducer Used	Computational Techniques	Publication Year/Reference
Gold nanorods (AuNR)-based	AuNR	PAI	Imaging of lymph vessels/nodes in breast cancer tumors	Phantom using PTFE tubes; in vivo on mice	Attenuation coefficient: −1.90 dB/mm380 times as compared ICG	Concave poly(vinylidene fluoride/trifluoroethylene) (P(VDF-TrFE)) US transducer	Delay-and-sum (DAS) beamforming method	2018 [154]
	^89^Zr-labeled bGNR@MSN(DOX)-PEG (Zirconium labeled PEGylated gold nanorods, GNR, coated with mesoporous silica nanoshell)	PAI, PET, PTT and chemotherapy	Imaging of 4T1 breast cancer tumors	In vitro and in vivo on mice	NP diameter: 135.9 nm; 4.7 fold stronger signal from PAI 24 h post-injection as compared to pre-injection	VEVO LAZR PA imaging system	-	2018 [155]
	AuNR coated with CTAB.	PAI, US	Imaging of tumor metastases in mice	In vivo EGFR-targeted PAI of lymph node metastases and tumor mass	Enhanced PA signal observed after 24 h in lymph node with metastases post-injection of gold nanorods.	LZ-550 linear array transducer, Vevo 2100 LAZR high-frequency US and PA imaging system.	-	2016 [127]
	Gold nanoparticles	PAI, US	Imaging of micro-metastases in lymph nodes	In vivo imaging of lymph node.	High spatial resolution images of micro-metastases (50 µm) were obtained after 2 h of peritumoral injection.	LZ-550 linear array transducer, Vevo LAZR high-frequency US and PA imaging system.	Spectral unmixing, sPA imaging algorithm to differentiate several optical absorbers.	2014 [156]
	Furin-cleavable RVRR (Arg-Val-Arg-Arg) peptides (Au-RRVR NPs)	PAI, PTT	Imaging HCT 116 colorectal carcinomas	In vitro and in vivo imaging of tumors	The PA signal reached an intensity maximum of approximately 8 h post-injection with a 1.6-fold enhancement compared to the initial background.	A multispectral optoacoustic tomography scanner with excitation light of 680–900 nm	Maynard operation sequence technique (MOST) measurement	2021 [157]
Gadolinium-/bismuth-based	Gd-PEG-Bi NPs (hydrophobic dodecanethiol-Bi nanoparticles, for CT and PA contrast, coated in gadolinium, for MRI, and PEG)	PAI, CT, MRI and for PTT	Imaging of C6 glial tumors	In vitro and in vivo on mice; hemolysis assay and in vivo blood clearance and bio-distribution	NP diameter: 45 nm; Strong PA signals at low concentrations of 0.625 mg/mL and after 30 min; Strongest PA signal at 3 h and blood half-life at 4.69 h; High biosafety and NIR absorption coefficient	Endra Nexus 128 PA imaging system	-	2018 [104]
Manganese-based	GO/MnWO_4_/PEG/DOX (Graphene-oxide, GO, grown in situ onto manganese tungsten oxide in the presence of PEG and loaded with doxorubicin)	PAI, MRI, PTT and chemotherapy	Imaging of breast cancer tumors (4T1 mouse mammary carcinoma)	In vitro and in vivo on mice; PTT, chemotherapy and cytotoxicity	Maximum PA signal observed at tumor region 6 h post-injection in vivo; however, the signal was maintained at 1.4 times that of pre-injection at 24 h. Little to no cytotoxicity observed	MOST inVision128, iThera Medical	-	2018 [105]
Iron oxide-based	Magnetic iron oxide nanoparticles	Molecular PAT	Imaging of 4T1 breast cancer tumors	In vivo molecular photoacoustic tomography of breast cancer in mice	Post injection of contrast agents PA signal increased 3 times after 5 min and 10 times after 24 h.	Focused-ultrasound transducer operating at 50 MHz and 3.5 MHz	Raster scanning to acquire PA images, Hilbert transform was used to process acquired signals.	2014 [158]
Copper(II) sulfide nanoparticles (CuS)	Copper(II) chloride, sodium sulfide, methoxy-PEG-thiol to form polyethylene glycol (PEG)-coated copper(II) sulfide nanoparticles	PAT	Imaging of 4T1 breast cancer tumors	In vivo PAT of blood vasculature of 4T1 breast cancer in mouse	After 2 h and 5 min of injecting contrast agent, PA signal had maximum intensity and minute details of blood vessels at tumor site were shown with great clarity.	-	-	2014 [159]

**Table 5 nanomaterials-12-00393-t005:** Classification and Summary of Tumor-Targeting In Vivo Studies for PAI Dye-based Contrast Agents.

Classification	Material Used	Imaging Modalities	Application	Studies Conducted	Relevant Measured Parameters	Transducer Used	Computational Techniques	Publication Year/Reference
ICG-based	ICG	PAI	Imaging of lymph vessels/nodes in breast cancer tumors	Phantom using PTFE tubes; in vivo on mice	Attenuation coefficient: −1.90 dB/mm	Concave poly(vinylidenefluoride/trifluoroethylene) (P(VDF-TrFE)) US transducer	Delay-and-sum (DAS) beamforming method	2018 [154]
	ICG-cRGD	PAI	Imaging of human glioblastoma (U-87MG, high αvβ3 expression) and epidermoid carcinoma (A431, low αvβ3 expression)	In vitro and in vivo on mice; followed by ex vivo of mice organs	Signal: plateaued after 30–60 min for ICG-RGD in U-87 MG and sustained for 24 h post-injection; 25 times greater for U-97MG than for A431	Vevo LAZR LZ250 PA imaging system	Spectral unmixing	2018 [17]
	SDF- 1/ICG/PFH/DOX PLGA NPs (PLGA shells encapsulating PFH, Doxorubicin and ICG and conjugated to chemokine SDF-1)	PAI, PTT and chemotherapy	Imaging of metastatic lymph nodes in tongue squamous cell carcinoma	In vitro and in vivo on rabbits	Signal: plateaued at 1 h and was sustained for 24 h post-injection; higher signal intensity for targeted groups than for non-targeted control	VEVO LAZR PA imaging system	-	2019 [160]
	Sodium hyaluronic acid, Ethylenediamine, ICG, single-walled carbon nanotubes	PAI	In vivo Imaging of SCC7 Tumor in mice	In vivo and ex vivo on mice	PA signal was not clear with the injection of free ICG. ICG combined with hyaluronic acid nanoparticles in SWCNT encapsulation provided strong signals. Image contrast decreased after 48 h of injecting IHANPT.	Endra Nexus128 imaging system	-	2016 [89]
	ICG, polyethylene glycol, reduced Nano-graphene oxide composite	PAI, Fluorescence imaging	In vivo imaging of Hela tumor (cervical carcinoma) models in mice	PAI of Phantoms, In Vivo PAI, In Vivo Toxicity Assessment	Nanocomposite produced minimal toxicity. Blood circulation time was 6 h. PAI showed accumulation and distribution of injected contrast agents at the tumor site.	Olympus focused ultrasound transducer with a central frequency of 10 MHz. Acoustic-resolution photoacoustic microscopy system	-	2016 [149]
Squaraine dye nanoprobe	squaraine dye SQ1 constructed from ethyl-grafted 1,8-naphtholactam and square acid in a donor-acceptor-donor structure	PAI, fluorescence imaging and PTT	PAI of breast cancer cells (MDA-MB-231 and MCF-7)	In vitro and in vivo imaging	SQ1nanoprobe performed well in both PA imaging and PTT of solid tumors.	PA images and corresponding PA intensities at 930 nm were obtained by a PA microscopy system	-	2020 [161]

### 2.5. Biosensors and Nanoprobes for In Vivo Tumor Studies

Since sensitivity and selectivity are vital for the early detection of cancer metastases, researchers have developed probes that react with molecules that have major functional roles in pathology—mainly in tumor growth. Such materials have been developed for other imaging modalities such as fluorescence imaging [162], PET, MRI, computed tomography (CT), and echography [163] and tested in vivo on tumor-bearing mice. PAI nanoprobes can detect cancer-associated growth on a molecular level and help clinicians administer precise dosage amounts to cancer patients, thus decreasing the adverse side effects of conventional chemotherapy. One such probe was developed by Wang et al. [78], where elevated nitric oxide levels, a characteristic of tumor sites, would lead to a linear increase in PA signal based on concentration levels. This probe was composed of benzothiadiazole conjugated to diphenylamine on both ends in a donor-acceptor-donor conformation. Chen et al. [146] used a novel approach to synthesize a pH-sensitive probe that performed the dual action of changing the PA signal intensity based on pH in the microenvironment, with increased signal in acidic environments, as well as performing photothermal action by self-assembly of human serum albumin and croconine-dye nanoparticles to eliminate tumor cells. The same research group used this probe to detect changes in tumor acidity by developing a metal, calcium ion and organic ligand, dicarboxylic cisplatin prodrug, nanoparticles covered with poly-l-histidine-PEG (pHis-PEG) to oxidize H_2_O_2_ and thus, perform dual-cancer therapy and real-time in vivo imaging [164].

There are a limited number of studies in the biosensor field, including PAI. This could be attributed to the tedious process of validating biosensors to determine whether they are clinically applicable and safe for use [165]. Since the interaction between the probe molecule and the target molecule should be highly selective, i.e., it should not respond adversely to other components in vivo, specifically, if other benign diseases exhibit increased concentrations of the target molecule. The reproducibility and commercialization of these biosensors might contribute to this comparatively unexplored field of probe and biosensor in cancer research. Figure 6 presents a summary of the studies conducted using different types of PAI contrast agents over the past four years.

## 3. Conclusions, Challenges and Future Directions

PAI is a promising modality for real-time imaging which can be developed by combining the properties of exogenous contrast agents such as metallic, carbon-based, semi-metallic, polymer-based nanomaterials and dyes. This review follows and categorizes the development of exogenous contrast agents for in vivo imaging, specifically tumor imaging. Different strategies have been exploited to enhance the PA signal, including incorporating moieties, inducing the plasmon coupling effect, coupling different classes of nanomaterials and dyes, forming stimuli-responsive nanoagents, magnetic and photo-induced electron transfer for background signal reduction, and self-assembly of materials. Other techniques include improving imaging systems for the synchronized detection of PA signals [166].

The field of optoacoustics/PA has vast implications for clinical translation, with one of the first studies for intraoperative breast cancer surgeries revealing successful tumor margin identification and identification of the biochemical contents. The current state-of-the-art in tumor margin definition is histological analysis, which requires the removal of the tumor and a 24-h period for analysis. The development of PAI implies that the clinical need for fast and real-time processing can finally be met in oncology. With regards to image characteristics, this study showed that PAI used a scan time of 20 min along with satisfactory penetration depths and high-resolution images compared to fluorescence, Raman spectroscopy, and diffuse reflection imaging [167]. Numerous studies revealed promising results [168]; however, extensive cohort studies are still needed before PAI can be used as a state-of-the-art imaging modality in oncology with a future trend towards NIR-II contrast agent development. Future studies may focus on developing more efficient stimuli-responsive agents that track shifts in absorptions peaks based on changes in the tumor microenvironment, while evaluating its toxicity, biocompatibility, biodegradability, and photostability.

## Figures and Tables

**Figure 1 nanomaterials-12-00393-f001:**
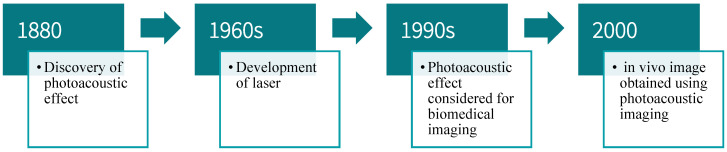
PAI timeline.

**Figure 2 nanomaterials-12-00393-f002:**
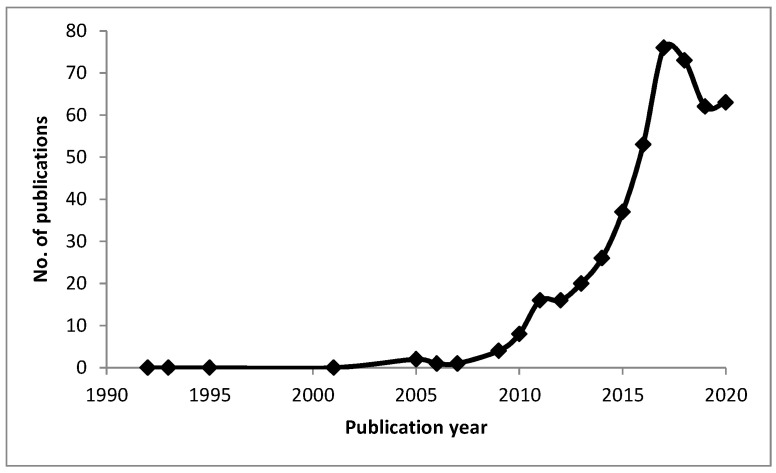
Emerging trend in the field of exogenous contrast agents for in vivo literature of tumors using photoacoustic imaging (Data compiled using Web of Science and ScienceDirect databases).

**Figure 3 nanomaterials-12-00393-f003:**
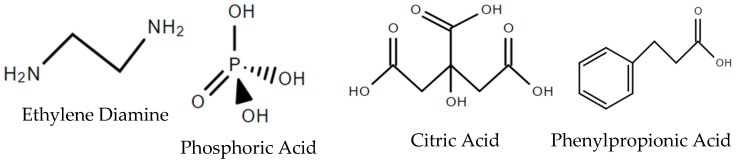
Chemical structures of natural sources for CDs’ production.

**Figure 4 nanomaterials-12-00393-f004:**
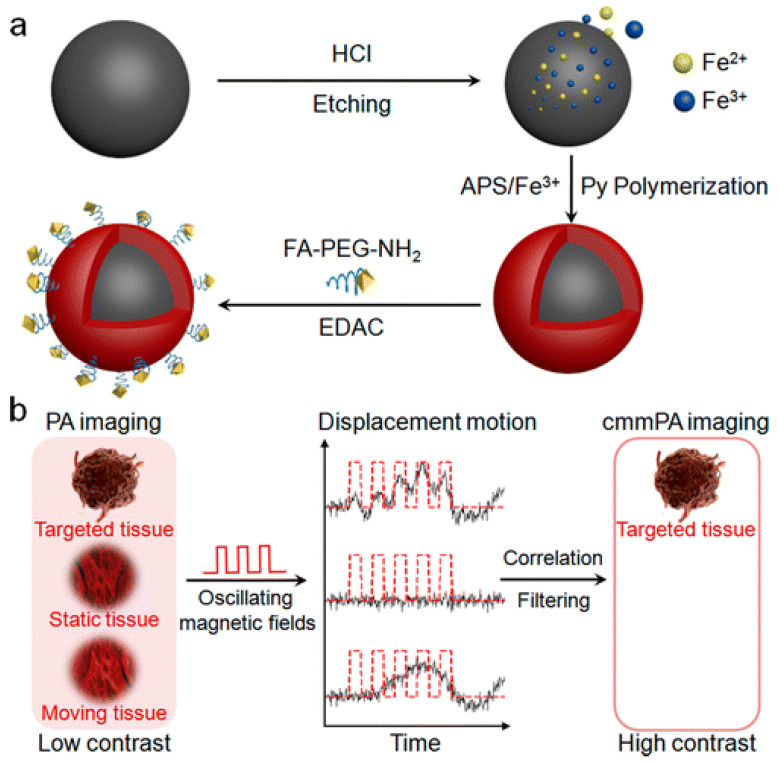
(**a**) Synthesis of MNP-PPy shell, a magneto-motive nanoparticle used for magnetic motion and speckle tracking (**b**) Distinctive displacement motion using magnetic fields created with synthesized magnetic nanoparticles within targeted tissue to eliminate PA background signal from non-ROI tissue [100].

**Figure 5 nanomaterials-12-00393-f005:**
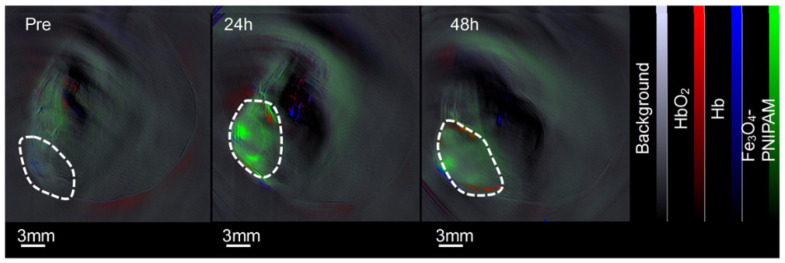
An example of the role contrast agents play in indicating tumor sites, with red representing oxyhemoglobin, blue representing deoxy-hemoglobin, and blue and green representing contrast agents [116].

**Figure 6 nanomaterials-12-00393-f006:**
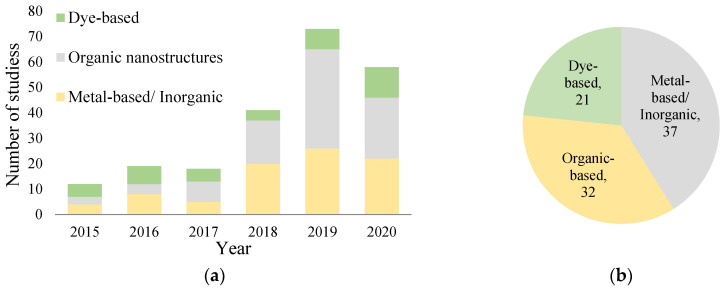
(**a**) No. of studies in the last four years. (**b**) Total number of studies in the last four years based on classification.

**Table 2 nanomaterials-12-00393-t002:** Detailed Parameters for Contrast Agent Studies.

Material	MCDs [50]	P-NP and N-NPs [56]	PGNR-PT6 and PGNR-PT7 [14]	MAGE-Au-PFH-NP [15]
Purpose	In vivo breast cancer imaging in mice	In vivo organ imaging in mice	In vivo osteosarcoma cancer imaging in mice	In vivo melanoma-tumor imaging in mice
No. of array elements	128 with a circular arc of 270° from 680 to 900 nm	256 with a circular arc of 270°	-	
Pulse duration (ns)	10	10	-	
Repetition rate (Hz)	10	10	-	
Central frequency (MHz)	5	5	-	21
Average size (nm)	40	20 for P-NP, and 100 for N-NP	81.7 for PGNR-PT6, and 82.7 for PGNR-PT7	354.27
Contrast	High	High	PGNR-PT6 and PGNR-PT7 enhanced contrast by 170% and 230%, respectively	High
Bio distribution (hours)	24	1 (more work needed)	24	24
Peak time (@Concentrationmax) (hours)	2	0.2	4	2
Biosafety	Low	Not measured	High	High
Physical efficacy	High	High	Very High	High

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
