# Peer review of "Exogenous Contrast Agents in Photoacoustic Imaging: An In Vivo Review for Tumor Imaging"

_nanomaterials, 2022, doi:10.3390/nano12030393_

Round 1
Reviewer 1 Report
From my point of view, the review is quite interesting and important. However, in order to have a bigger impact on society authors should also make a link between contrast agents and techniques well adopted for the specific types of the particles.
Additionally, some pictures demonstrating the efficiency of specific contrast should be provided.
In line 267 there is a problem with linking.
Author Response
Dear Respected Reviewer,
We would like to thank you for your valuable comments and suggestions. The manuscript has been revised and benefited substantially from these insightful suggestions, for which we are grateful. We have made every attempt to accommodate your comments, and we have hopefully addressed all of them satisfactorily.
Point-by-point replies to all comments have been provided in the attached document and the manuscript has been revised accordingly with changes tracked as requested.
Best regards,
Authors

Reviewer 2 Report
Manuscript reference nanomaterials-1530069
Title: Exogenous Contrast Agents in Photoacoustic Imaging: An in Vivo Review for Tumor Imaging
by Afifa Farooq et al.
The authors theoretically studied resonance effects in rectangular dielectric surface-relief gratings, illuminated with a limited cross-section Gaussian Beam. The paper is very interesting, well written, and could be published after some major revisions so to address the following critical points:
a) The authors used Section 1 to introduce and explain the abbreviations, but this seems to differ from the policy of the journal: Acronyms/Abbreviations/Initialisms should be defined the first time they appear in each of three sections: the abstract; the main text; the first figure or table. When defined for the first time, the acronym/abbreviation/initialism should be added in parentheses after the written-out form. We guess that the author should follow these suggestions.
b) Line 60: “refer to Error! Reference source not found”. The authors should correct the error.
c) Line 72: “PAI [21–29] (See Error! Reference source not found.)”. The authors should correct the error.
d) Line 172: “(refer Error! Reference source not found.) [45]”. The authors should correct the error.
e) Line 267: “(See Error! Reference source not found.).”. The authors should correct the error.
f) Line 211: “[51,52,61–70,53,71–74,54–60]”. The citations should be put in a sequential order. The same happens again at Line 299 “[58,59,94–96,60,62,84,89–93] (2018), [88,97–100] (2017), [101–107] (2016), 299 [85,108–110] (2015)”
g) Line 235: “With an encapsulation efficiency of 70.61%”. Accuracy in the evaluation of the efficiency seems too high and not realistic.
h) Line 271: “This imaging technique has the ability to detect tumor cells in the 100s – 1000s cell range”: this sentence is not clear. The authors should explain more in details the 100s – 1000s cell range.
i) Table 1: there are several explanations to be added:
“No. of array elements 128 covering 270⁰ 256 covering 270⁰”- specify what is 270°
“Average size (nm) 40 20 -100 81.7, 82.7 354.27” - which one is the right number? 81.7 or 82.7?
“Contrast High High >170% & 230% High” - what do the authors mean for 170% & 230%?
j) Table 2: this table occupies 5 pages, bringing information in a chaotic way. We suggest the authors to organize the useful contents in a different way, perhaps by using additional tables.
k) We guess that the authors should enrich the bibliography by citing in the introduction some additional review papers on the development of photoacoustics as for example
[1] M. Bertolotti and R. Li Voti, A note on the history of photoacoustic, thermal lensing, and photothermal deflection techniques, Journal of Applied Physics 128, 230901 (2020)
l) we guess that in the introduction or better in Section 3.3 (Metal/ Inorganic Contrast Agents) it is important to mention that there are other techniques allowing to retrieve individual contribution of scattering and absorption by applying for example photoacoustic techniques in frequency regime for the detection of the statistics of metal NPs and their aggregates as recently shown in the following Refs. to be quoted
[2] Li Voti R. et al Nanoscale Advances 3, Pages 4692 - 470121 (2021)
[3] F.R. Lamastra, et al, Nanotechnology 28, 375704 (2017)
Author Response

(The authors gave the same response as above.)

Reviewer 3 Report
This manuscript summarizes some exogenous contrast agents for photoacoustic imaging of tumor. This paper details the different types of contrast agents used in this diagnostic field, i.e., organic-based, metal/inorganic-based, and dye-based contrast agents. Also, the discussion of the challenges and future direction of PA imaging are presented. This paper can be accepted after major revisions.
- The paragraph beginning from line 136: “PA imaging offers several advantages, such as…”, this introduction on PA imaging is superfluous here. The PAI introduction including advantages, working mechanism, and features, should be described earlier.
- Organic nanomaterials have excellent properties for in vivo imaging. However, the contents in this section are too short. Authors should classify some representative organic probes for PA imaging, such as small organic dyes, conjugated polymers or oligomers…
- NIR-II PA imaging exhibits more promising than NIR-I PA imaging in terms of imaging depth, resolution, and sensitivity. Authors should include this important content in the manuscript.
Author Response

(The authors gave the same response as above.)

Round 2
Reviewer 2 Report
The authors revised the manuscript according to the referee suggestions. The manuscript is now much more imporved after revisions and can be published on the journal
Reviewer 3 Report
The authors have addressed all my concerns, and the manuscript can be accepted for publication now.